# Distant Nodes Seen on PSMA PET-CT Staging Predicts Post-Treatment Progression in Men with Newly Diagnosed Prostate Cancer—A Prospective Cohort Study

**DOI:** 10.3390/cancers14246134

**Published:** 2022-12-13

**Authors:** Sean Ong, Claire Pascoe, Brian D. Kelly, Zita Ballok, David Webb, Damien Bolton, Declan Murphy, Shomik Sengupta, Patrick Bowden, Nathan Lawrentschuk

**Affiliations:** 1EJ Whitten Foundation Prostate Cancer Research Centre, Epworth HealthCare, Richmond, VIC 3121, Australia; 2Young Urology Researcher’s Organisation, Melbourne, VIC 3000, Australia; 3Department of Surgery, University of Melbourne, Parkville, VIC 3010, Australia; 4Division of Cancer Surgery, Peter MacCallum Cancer Centre, Melbourne, VIC 3000, Australia; 5Department of Urology, Eastern Health, Box Hill, VIC 3128, Australia; 6Department of Nuclear Medicine, Richmond Medical Imaging, Richmond, VIC 3121, Australia; 7Olivia Newton-John Cancer and Wellness Centre, Austin Health, Heidelberg, VIC 3084, Australia; 8Eastern Health Clinical School, Monash University, Box Hill, VIC 3128, Australia; 9Department of Urology, Royal Melbourne Hospital, Melbourne, VIC 3051, Australia

**Keywords:** prostate cancer, PSMA PET, primary staging

## Abstract

**Simple Summary:**

PSMA PET-CT is a new scan which has now been proven to be the most accurate in detecting prostate cancer outside the prostate gland. It is now recommended by international guidelines to be used before treatment is commenced for prostate cancer. However, we still do not know the true impact the information from these scans has on the treatment that we give the patients. In this study, we look at whether men who had a PSMA PET-CT scans before their treatment for prostate cancer needed any further treatment for their prostate cancer within the first 29 months of follow up. Our results found that 80% of men these men did not need any further treatment for their prostate cancer. We also found that if the prostate cancer was seen on PSMA PET-CT in lymph nodes outside of the pelvis, men were 5 times more likely to need another form of treatment within 29 months.

**Abstract:**

PSMA PET-CT scans are now recommended in international urological guidelines for primary staging and re-staging of prostate cancer. However, there is little published literature on the clinical outcomes for patients after treatment decisions made using PSMA PET-CT results. This is a multisite, prospective cohort study investigating the clinical outcomes of men who received treatment plans based on PSMA PET-CT results for primary staging. Men with biopsy proven prostate cancer received a PSMA PET-CT scan for primary staging. Treatment plans were recommended by multidisciplinary teams (MDT). After treatment, these men were followed with 6 monthly PSA tests and imaging or biopsies if recommended by MDT. The primary outcome was treatment progression defined as the addition or change of any treatment modalities such as androgen deprivation therapy, radiation therapy or chemotherapy. In total, 80% of men did not have any treatment progression after enactment of treatment based on PSMA PET-CT primary staging results at 29 months of follow up. Men who had distant nodes seen on PSMA PET-CT had a 5 times increased risk of treatment progression. Larger studies with longer follow up are needed to validate our results and optimise the way clinicians use PSMA PET-CT results to guide management.

## 1. Introduction

Prostate specific membrane antigen (PSMA) is a cell surface transmembrane glycoprotein that is overexpressed in prostate cancer (PCa) cells relative to physiological levels in the small intestine, kidney and salivary glands [1]. Studies have found that PSMA is expressed in over 90% of PCa cells at rates that are 100–1000 times greater than physiological levels especially in higher grade cancers [2,3,4]. Thus, PSMA provides a target for diagnostic imaging and therapeutic advances. For PCa detection, small molecules such as gallium (^68^Ga) or fluorine (^18^F) with an attached radio-labelled marker can be used to target PSMA [5]. This is combined with positron emission tomography (PET) technology to produce an image that can identify the presence of PCa [6].

The introduction of PSMA PET based imaging has reshaped the diagnostic and therapeutic landscape for PCa. PSMA PET-CT (computed tomography) is now recommended in international urological guidelines [7] for primary staging and re-staging of PCa due to results demonstrating superior accuracy and less radiation compared to conventional imaging [8,9]. Furthermore, PSMA PET-CT also has the potential to save costs to healthcare systems [10]. Despite this, some are still wary of its use given the lack of evidence in the current literature on PSMA PET-CT’s impact upon the clinical outcomes of patients.

Nevertheless, the published evidence so far on PSMA PET-CT has culminated in the increased availability and accessibility of this scan around the world. An example of this is Australia, where the inclusion of PSMA PET-CT in urological guidelines has culminated in a government rebated PSMA PET-CT scan for men who have histologically confirmed intermediate or high risk PCa or biochemical recurrent PCa [11,12]. Given its inevitable increase in utility, it is important to determine if treatment decisions made using PSMA PET-CT staging scans has a positive impact on the clinical outcomes of patients. Prostate cancer specialists should be able to use information from PSMA PET-CT results to optimise the treatment of men with PCa.

In this prospective cohort study, we aim to report and analyse the clinical outcomes of men who received management plans based on PSMA PET-CT results for primary staging of their PCa. Furthermore, we seek to identify any predictors of outcomes to optimise treatment strategy for patients.

## 2. Materials and Methods

### 2.1. Ethics

Ethical approval was obtained from the Monash Health Human Research and Ethics Committee (HREC Number: RES-19-0000-874E).

### 2.2. Patients

Patients were recruited through the seven campuses of Epworth HealthCare hospitals throughout Victoria, Australia between March 2017 and March 2018. All patients signed an information and consent form to be recruited into the study.

### 2.3. Primary Endpoint

The primary endpoint of this study was treatment progression post initial management plan. This was defined as the addition or change of any treatment modalities such as androgen deprivation therapy (ADT), radiation therapy or chemotherapy. Disease progression seen on imaging or as a PSA progression with no addition or change in treatment modality was not classified as treatment progression.

### 2.4. Inclusion and Exclusion Criteria

Men were eligible for the study if they had histologically proven intermediate (PSA 10–20 ng/mL, Gleason = 7 (4 + 3), clinical (or MRI) T2-T3) or high risk (PSA > 20 ng/mL, Gleason ≥ 8, clinical (or MRI) ≥ T3 (extraprostatic extension)) PCa and were over 18 years of age. Patients were recruited irrespective of conventional imaging findings.

Exclusion criteria was another active malignancy within the last 5 years, excluding non-melanoma skin cancer. Patients were also excluded if they had undergone any form of treatment for his disease (including, but not limited to, surgery, radiotherapy, brachytherapy or hormone therapy).

### 2.5. Study Design

This study was a multi-centre prospective cohort study of men with newly diagnosed intermediate or high risk PCa based on the inclusion and exclusion criteria mentioned above.

Patients were enrolled and consented for the study. Baseline demographic data was collected along with medical history regarding their prostate cancer diagnosis and prior treatment. These men then received a PSMA PET-CT scan as part of their cancer staging. Each case including the results of the PSMA PET-CT scan were then presented in a multidisciplinary meeting (MDM) including Urologists, Medical Oncologists, Nuclear Medicine Physicians, Radiation Oncologists, Uro-pathologists and Uro-radiologists to determine a consensus management plan. Men received a minimum of 6 monthly PSA tests for follow up with imaging or further presentation to an MDM based on PSA results or clinical concern. The need for treatment progression and type of treatment was discussed in MDM consensus. Data pertaining to follow up investigations, and treatment additions or changes were prospectively collected until Mar 2021. 

### 2.6. PSMA PET-CT Imaging and Radiotracer

PSMA PET-CT scans were performed at various radiographic centres around Victoria, Australia. All centres used ^68^Gallium-PSMA-11 as the radiotracer. These were produced using a standardised protocol which defined minimum specifications for radiopharmaceutical production. ^68^Gallium-PSMA-11 scanning protocols were similar across sites. PET-CT scanners acquired PET images from thighs to vertex at 60 min after administration of 2 MBq/kg body weight ± 5% of ^68^Gallium-PSMA-11. A low dose non-contrast CT was performed during tidal respiration for attenuation correction and anatomical correlation.

All PSMA PET-CT scans were reported by a radiologist at the centre where they were performed and then reviewed by a nuclear medicine specialist at a multi-disciplinary meeting. Lesions were assessed as positive by the nuclear medicine specialist.

### 2.7. Statistical Analysis

Univariate and multivariate logistic regression analysis was performed. All statistical analysis was performed using R software. A *p* value of <0.05 was considered to be statistically significant.

## 3. Results

### 3.1. Patient Characteristics

In total, 84 men were recruited into this study. The median age of these men is 73 years and median PSA at time of PSMA PET was 10 ng/mL. Prostate MRI was performed on 54 out of the 86 men. PIRADS 2, 3, 4, 5 lesions were found in 6, 1, 17, 30 men, respectively. ISUP grade 1, 2, 3, 4, 5 PCa was found on prostate biopsy for 1,16, 36, 18 and 13 men, respectively. The histological grade of PCa on biopsy for 2 men was unknown. This is summarised in Table 1.

### 3.2. PSMA PET-CT Scan Results

PSMA PET scans revealed avid prostate lesions in 67 men, avid pelvic lymph nodes in 23 men and avid extra-pelvic lymph nodes in 10 men. Bony lesions were seen in 13 men and a thyroid lesion was seen on 1 scan. This is seen in Table 2.

### 3.3. Management Decisions

Radical prostatectomy and external beam radiation therapy was the chosen management plan for 48 and 25 men, respectively. A total of 4 men received androgen deprivation therapy only and 9 men were planned for chemotherapy. Table 3 summarises this. 

### 3.4. Follow up

Six of the 86 men were lost to follow up. Of the remaining men, the mean follow up time was 29 months (4–47 months) post PSMA PET-CT scan. On follow up, 64 (64/80: 80%) men had no treatment progression. However, 14 (14/78: 18%) men progressed to received further treatment. Progression to androgen deprivation therapy only, radiation therapy and chemotherapy was observed in 6, 5 and 5 patients, respectively. Table 4 and Table 5 summarises the follow up data.

### 3.5. Predictors of Treatment Progression

A univariate analysis on each variable was performed to ascertain any predictors of treatment progression. Only avid distant nodes (outside the pelvis) found on PSMA PET-CT was a statistically significant predictor of treatment progression (odds ratio (OR) 5.36, 95% CI 1.30–22.54, *p* = 0.018). A negative PSMA PET-CT was not a statistically significant predictor of no treatment progression. On multivariate analysis controlling for age, PSA and grade group on prostate biopsy, presence of distant nodes on PSMA PET-CT remained a statistically significant predictor of treatment progression (Table 6). 

## 4. Discussion

PSMA PET-CT is now embedded in urological guidelines for primary staging of men with high risk PCa [7]. It has been shown to have greater detection rates of metastases and less radiation compared to conventional CT and whole body bone scan [8]. Subsequent management informed by PSMA PET imaging has not been widely investigated [13]. With the more accurate PSMA PET-CT comes the detection of lesions not seen previously on conventional imaging. It is therefore important to ensure that clinical management decisions made with this new information is leads to good long term outcomes for patients [7]. Furthermore, can PSMA PET-CT be utilised as a biomarker to predict outcomes and therefore guide management? 

In this multisite prospective cohort study of 86 men, we reported the medium term outcomes of men with PCa who received clinical management plans using PSMA PET-CT primary staging. At median follow up of 29 months, our results showed that 80% of men did not have any treatment progression which was defined as the addition or change of any treatment modalities such as androgen deprivation therapy (ADT), radiation therapy or chemotherapy. 

Distant nodes seen on PSMA PET-CT was found to be the only statistically significant predictor of treatment progression. These findings could imply that the detection of distant nodes allowed intensification of upfront therapy (chemotherapy or ARPIs) which the literature shows has benefit [14]. Interestingly, a negative PSMA PET-CT was not a statistically significant predictor of no treatment progression which may indicate the non-visible micro-metastatic disease leading to treatment progression. Ultimately, further studies are needed to optimise the selection of men for treatment using PSMA PET-CT findings.

The literature has limited data investigating outcomes for patients post management decisions made with primary PSMA PET-CT staging. Roberts et al. [15]. investigated a cohort of 71 men with biopsy proven PCa staged with PSMA PET-CT prior to radical prostatectomy. They reported that 53/71 (74.6%) of men had no progression (defined as biochemical recurrence (post-operative serum PSA > 0.2 ng/mL on two separate occasions at minimum 2-week intervals) OR commencement or referral for adjuvant or salvage therapies, such as radiotherapy or androgen deprivation therapy, in the absence of biochemical recurrence) at a median follow up of 19.5 months. Interestingly, our results were similar despite our cohort of men being both localised and metastatic, undergoing multiple modalities of treatment (both localised and systemic treatments) and having longer follow up (29 vs. 19.5 months). This is probably because our cohort contained only a limited number of men with metastatic PCa, the majority being prostate confined. In fact, in men with only prostate confined disease found on PSMA PET-CT, 87% did not have any treatment progression. 

Clearly, the majority of men with treatment progression in our study had metastatic disease. The treatment strategy for men with metastatic deposits seen on PSMA PET-CT remains a contentious issue due to most studies being based on conventional imaging. This is particularly apparent in low volume metastatic disease where the decision to offer systemic therapies or aggressive metastatic directed therapy remains challenging. Further studies focussing on this cohort of patients are needed to optimise treatment in this setting. 

PSMA PET-CT as a prognostic biomarker has also been investigated however mostly in the biochemical recurrent and metastatic PCa setting. Roberts et al. found intraprostatic PSMA intensity was predictive of progression [15]. Two studies showed that a negative PSMA PET-CT scan in the biochemical recurrent setting is predictive of progression at 3 years follow up [16,17]. Our results found that men who had distant nodes seen on PSMA PET-CT had a 5-fold increase in having treatment progression. This is probably reflective of an aggressive form of metastatic PCa having spread outside the pelvis or may suggest that these men may need more optimal treatment for their disease. Our results should provoke further studies investigating the use of PSMA PET-CT for risk stratification to identify the optimal treatment regimes for newly diagnosed metastatic PCa. This is particularly pertinent given the introduction of duplet and triplet combination therapies using ADT, chemotherapy and novel androgen receptor pathway inhibitors for metastatic hormone sensitive PCa [18]. Many risk classifications made using conventional imaging need updating in the PSMA PET-CT era. PSMA PET-CT has already been utilised as a prognostic marker to guide treatment in the metastatic castrate-resistant PCa setting [19].

There are several limitations of this study. We analysed a small heterogenous cohort of men that underwent varying treatments for their PCa. Their cases were discussed in multiple different multidisciplinary team meetings and patient preference was factored into the clinical decision-making process for all men. However, this represents a real-world scenario for clinical decision making. The different treatment modalities given to these men may have skewed results and in a future study choosing a specific cohort to analyse may give more meaningful results. Nevertheless, we hope that our study is a catalyst for future trials in this area with larger sample sizes and longer follow up. 

## 5. Conclusions

In conclusion, this multisite prospective cohort study found that 80% of men did not have any treatment progression after enactment of treatment based on PSMA PET-CT primary staging results at 29 months of follow up. Men who had distant nodes seen on PSMA PET-CT had a 5 fold increase in risk of treatment progression. Larger studies with longer follow up are needed to validate our results and optimise the way clinicians use PSMA PET-CT results to guide management.

## Figures and Tables

**Table 1 cancers-14-06134-t001:** Patient characteristics.

Number of patients recruited		86
Mean age		73 years
Median PSA at time of PSMA PET-CT		10 ng/mL (IQR 5.9–20)
MRI received	Yes	54
	No	32
MRI findings	PIRADS 2	6
	PIRADS 3	1
	PIRADS 4	17
	PIRADS 5	30
ISUP grade	Unknown	2
	ISUP grade 1	1
	ISUP grade 2	16
	ISUP grade 3	36
	ISUP grade 4	18
	ISUP grade 5	13

**Table 2 cancers-14-06134-t002:** PSMA PET-CT results.

Positive	84
Negative	2
Prostate lesion avid	68
Pelvic lymph nodes	23
Extra-pelvic lymph nodes	10
Bony lesion	13
Distant organ lesion	1

**Table 3 cancers-14-06134-t003:** Management decisions.

Radical Prostatectomy	48
Radiation Therapy	25
Androgen Deprivation Therapy only	4
Chemotherapy or Androgen Receptor Pathway Inhibitor	9

**Table 4 cancers-14-06134-t004:** Follow up.

Lost to follow up	6
Mean follow up	29 months (4–47 months)
No treatment progression	64 (80%)
Treatment progression	16 (18%)
Progression on imaging but no intervention	2 (2.5%)
Progressed to ADT	6 (7.5%)
Progressed to RTx	5 (6.25%)
Progressed to chemotherapy or ARPI	5 (6.25%)

**Table 5 cancers-14-06134-t005:** No treatment progression vs. treatment progression.

	Treatment Progression	No Treatment Progression
Total	16	64
Mean PSA	54 ng/mL	22 ng/mL
ISUP grade 1	0 (0%)	1 (1.56%)
ISUP grade 2	1 (6.25%)	12 (18.75%)
ISUP grade 3	5 (31%)	29 (45.3%)
ISUP grade 4	3 (18.75%)	15 (23.43%)
ISUP grade 5	7 (43.75%)	5 (7.8%)
PSMA PET-CT showed avid lesion outside prostate	9 (32%)	19 (68%)
PSMA PET-CT showed prostate confined disease	7 (13%)	45 (87%)

**Table 6 cancers-14-06134-t006:** Multivariate analysis of variables predicting treatment progression.

Predictors	Odds Ratio	95% Confidence Interval	*p*-Value
Age	0.97	0.90–1.04	0.357
Serum PSA	1.00	1.00–1.01	0.254
ISUP grade on prostate biopsy	0.99	NA–1.03	0.694
Distant nodes PSMA PET-CT	5.30	1.25–23.09	0.022

## Data Availability

The datasets used and/or analysed during the current study are available from the corresponding author on reasonable request.

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
