# Peer review of "Distant Nodes Seen on PSMA PET-CT Staging Predicts Post-Treatment Progression in Men with Newly Diagnosed Prostate Cancer—A Prospective Cohort Study"

_cancers, 2022, doi:10.3390/cancers14246134_

Round 1
Reviewer 1 Report
This study tried to show the true impact of PSMA PET-CT scans for primary staging of PCa. Their results found that 80% of men these men did not need any further treatment for their PCa. Although the content of this paper is very interesting, it is fraught with major problems.
The number of cases is too small to discuss the prognosis of prostate cancer, the follow-up period is short. Furthermore, T stage was unknown because MRI was not performed in all cases, and the risk classification of the target cases was not performed.
Author Response
Thank you for your comments and time reviewing our manuscript. Although we agree our case number is small and follow up is not long term, our paper should provoke further studies in this area as PSMA PET-CT becomes more commonly utilised around world.
In regards to MRI, many of these cases were done in an era before MRI was subsidised for patients in Australia. Indeed MRI is still not routine in many countries such as the USA. Every man however had prostate cancer confirmed on biopsy with extensive sampling and although grade of cancer may have been different with MRI targeting, we do not think management plans would have been significantly different.
Reviewer 2 Report
This reviewer appreciates the authors’ work on this field. Though this work is interesting there is several places where the manuscript should be improved prior to acceptance.
1. Various grammatical errors throughout, line 35 for example.
2. How did the population of those imaged with PSMA PET-CT compared to a similar cohort not imaged? Was this similar with 80% not needing/receiving further treatment? The authors suggest that there is a lack of evidence in PSMA PET-CT’s impact on clinical outcome and comparing outcomes of PSMA PET-CT patients to others would assist in that clinical gap.
3. Did imaging improve the selection of patients receiving additional therapy?
4. Further details need to be added to the study to indicate how this study adds to the field? Otherwise, this reviewer has a hard time understanding how this is different than previous studies showing PSMA imaging’s purpose when it is already the international recommendation.
5. Was there any analysis performed on the differences in what the treatment/management decisions were with the added information from the PSMA PET-CT compared to what would have been proposed if imaging was not included in the course of care.
6. The authors report that a negative PSMA PET-CT scan was not a predictor of no treatment progression. The authors could comment on this in the discussion. Could this lead to over/under treatment?
Author Response
Thank you for your comments and time reviewing our manuscript. Below are our responses.
Comment 1: Various grammatical errors throughout, line 35 for example.
Response 1: Thank you, we have reviewed the manuscript and corrected grammatical errors.
Comment 2: How did the population of those imaged with PSMA PET-CT compared to a similar cohort not imaged? Was this similar with 80% not needing/receiving further treatment? The authors suggest that there is a lack of evidence in PSMA PET-CT’s impact on clinical outcome and comparing outcomes of PSMA PET-CT patients to others would assist in that clinical gap.
Response 2: Thank you, we agree that a comparison to a cohort not imaged would be helpful however we could not find any studies with men who were not imaged or a similar outcome in men with who used conventional imaging. It is also difficult to compare to this cohort as it contained a heterogenous group of men (localised, locally advanced and metastatic disease).
We have compared in our manuscript to a similar study using PSMA PET-CT for localised disease only and 2 studies in the biochemical recurrent setting. These were the most relevant comparisons we could find.
Comment 3: Did imaging improve the selection of patients receiving additional therapy?
Response 3: Unfortunately, this question is beyond the scope of this study as there was no comparison group and our cohort was heterogenous (localised, locally advanced and metastatic disease). We have however mentioned the following in our manuscript to emphasise the need to investigate the selection of patient receiving additional therapy based on PSMA PET.
“Ultimately, further studies are needed to optimise the selection of men for treatment using PSMA PET-CT findings.”
Comment 4: Further details need to be added to the study to indicate how this study adds to the field? Otherwise, this reviewer has a hard time understanding how this is different than previous studies showing PSMA imaging’s purpose when it is already the international recommendation.
Response 4: Although PSMA PET-CT is mentioned in international guidelines, the same guidelines also caution against the use of PSMA PET-CT because of the lack of evidence in regards to outcomes after management using PSMA results. We mention this in our introduction “Despite this, some are still wary of its use given the lack of evidence in the current literature on PSMA PET-CT’s impact upon the clinical outcomes of patients.”
Ultimately, the inevitable move away from conventional CT/WBBS to PSMA PET-CT will yield new results which clinicians will have to learn to utilise to optimise clinical management. We mention this in out conclusion. “Larger studies with longer follow up are needed to validate our results and optimise the way clinicians use PSMA PET-CT results to guide management.”
Comment 5: Was there any analysis performed on the differences in what the treatment/management decisions were with the added information from the PSMA PET-CT compared to what would have been proposed if imaging was not included in the course of care.
Response 5: Unfortunately not. This would have been best compared to a group receiving conventional imaging (CT/WBBS) but unfortunately, most men received PSMA PET-CT only. MDM outcomes were only reported using PSMA PET-CT results. Moreover, the proPSMA study has reported this difference in their study. We focussed more on the outcomes from these decisions.
Comment 6: The authors report that a negative PSMA PET-CT scan was not a predictor of no treatment progression. The authors could comment on this in the discussion. Could this lead to over/under treatment?
Response 6: Thank you, we have now added this paragraph to the manuscript.
“Distant nodes seen on PSMA PET-CT was found to be the only statistically significant predictor of treatment progression. These findings could imply that the detection of distant nodes allowed intensification of upfront therapy (chemotherapy or ARPIs) which the literature shows has benefit[14]. Interestingly, a negative PSMA PET-CT was not a statistically significant predictor of no treatment progression which may indicate the non-visible micro-metastatic disease leading to treatment progression.”
Round 2
Reviewer 1 Report
The manuscript has been revised well. I think this manuscript will be acceptable.
Author Response
Thank you so much for your time and comments
Reviewer 2 Report
Not all additions mentioned in the response letter are shown as track changes in the manuscript. Was that purposeful? For example, response 3 is not a quote from the manuscript. Response 4 does not match the introduction changes in paragraphs 1 and 2.
Author Response
Thank you again for your comments. Sorry we should have made our response clearer.
Please find quote from response 3 on page 7 line 210-212 (discussion section). This is highlighted as tracked changes.
The 2 quotes from response 4 were already present in the original manuscript. Page 2 line 64 (introduction) and Page 8 line 264 (conclusion).
Round 3
Reviewer 2 Report
ok